# Contrastive Unsupervised Learning of World Model with Invariant Causal Features

## Abstract

In this paper we present a *world model*, which learns causal features using the invariance principle. In particular, we use contrastive unsupervised learning to learn the invariant causal features, which enforces invariance across augmentations of irrelevant parts or styles of the observation. The world-model-based reinforcement learning methods independently optimize representation learning and the policy. Thus naïve contrastive loss implementation collapses due to a lack of supervisory signals to the representation learning module. We propose an intervention invariant auxiliary task to mitigate this issue. Specifically, we use data augmentation as style intervention on the RGB observation space and depth prediction as an auxiliary task to explicitly enforce the invariance. Our proposed method significantly outperforms current state-of-the-art model-based and model-free reinforcement learning methods on out-of-distribution point navigation tasks on the iGibson dataset. Moreover, our proposed model excels at the sim-to-real transfer of our perception learning module. Finally, we evaluate our approach on the DeepMind control suite and enforce invariance only implicitly since depth is not available. Nevertheless, our proposed model performs on par with the state-of-the-art counterpart.

## 1 Introduction

An important branching point in reinforcement learning (RL) methods is whether the agent learns with or without a predictive environment model. In model-based methods, an explicit predictive model of the world is learned, enabling the agent to plan by thinking ahead (Deisenroth & Rasmussen, 2011; Silver et al., 2018; Ha & Schmidhuber, 2018; Hafner et al., 2021). The alternative model-free methods do not learn the predictive model of the environment explicitly as the control policy is learned end-to-end from the pixels. As a consequence, model-free methods do not consider the future downstream tasks. Therefore, we hope that model-based methods are more suitable for out-of-distribution (OoD) generalization and sim-to-real transfer.

A model-based approach has to learn the model of the environment purely from experience, which poses several challenges. The main problem is the training bias in the model, which can be exploited by an agent and lead to poor performance during testing (Ha & Schmidhuber, 2018). Further, model-based RL methods learn the representation using observation reconstruction loss, for example variational autoencoders (VAE) (Kingma & Welling, 2014). The downside of such a state abstraction method is that it is not suited to separate the task relevant states from irrelevant ones, resulting in current RL algorithms often overfit to environment-specific characteristics Zhang et al. (2020). Hence, relevant state abstraction is essential for robust RL model, which is the aim of this paper.

Causality is the study of learning cause and effect relationships. Learning causality in pixel-based control involves two tasks. The first is a causal variable abstraction from images, and the second is learning the causal structure. Causal inference uses graphical modelling (Lauritzen & Spiegelhalter, 1988), structural equation modelling (Bollen, 1989), or counterfactuals (Dawid, 2000). Pearl (2009) provided an excellent overview of those methods. However, in complex visual control tasks the number of state variables involved is high, so inference of the underlying causal structure of the model becomes intractable (Peters et al., 2016) . Causal discovery using the invariance principle tries to overcome this issue and is therefore gaining attention in the literature (Peters et al., 2016; Arjovsky

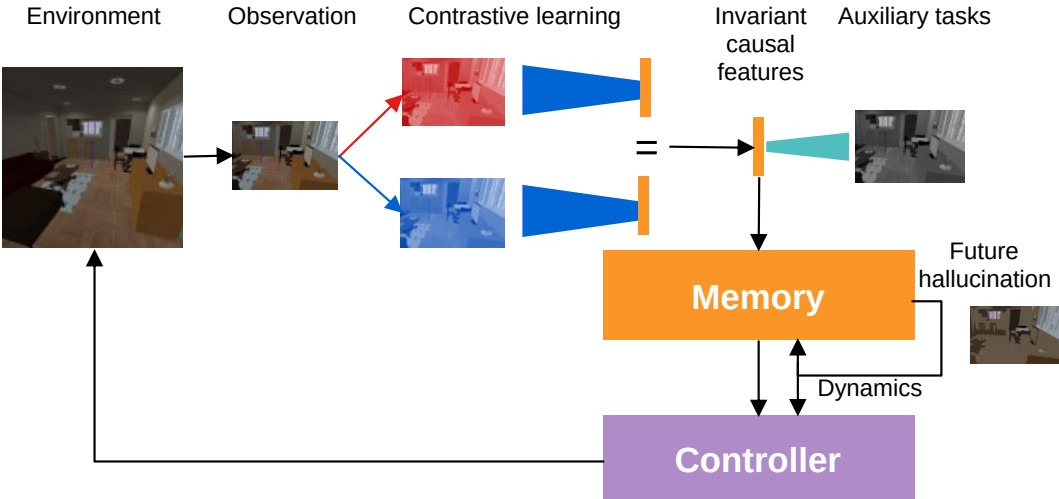

Figure 1: Flow diagram of proposed *World Model with invariant Causal features* (WMC). It consists of three components: i) unsupervised causal representation learning, ii) memory, and iii) controller.

et al., 2020; Zhang et al., 2020; Mitrovic et al., 2021). Arjovsky et al. (2020) learns robust classifiers based on invariant causal associations between variables from different environments. Zhang et al. (2020) uses multiple environments to learn the invariant causal features using a common encoder. Here spurious or irrelevant features are learnt using environment specific encoders. However, these methods need multiple sources of environments with specific interventions or variations. In contrast, we propose using data augmentation as a source of intervention, where samples can come from as little as a single environment, and we use contrastive learning for invariant feature abstraction. Related to our work, Mitrovic et al. (2021) proposed a regularizer for self-supervised contrastive learning. On the other hand, we propose an intervention invariant auxiliary task for robust feature learning.

Model-based RL methods do not learn the feature and control policy together to prevent the greedy feature learning. The aim is that the features of model-based RL will be more useful for various downstream tasks. Hence, state abstraction uses reward prediction, reconstruction loss or both (Ha & Schmidhuber, 2018; Zhang et al., 2020; Hafner et al., 2021). On the other hand contrastive learning does not use the reconstruction of the inputs and applies the loss at the embedding space. Therefore, we propose a causally invariant auxiliary task for invariant causal features learning. Specifically, we utilize depth predictions to extract the geometrical features needed for navigation, which are not dependent on the texture. Finally, we emphasize that depth is not required for deployment, enabling wider applicability of the proposed model. Importantly, our setup allows us to use popular contrastive learning on model-based RL methods and improves the sample efficiency and the OoD generalization.

In summary, we propose a *World Model with invariant Causal features* (WMC), which can extract and predict the causal features (Figure 1). Our WMC is verified on the *point goal* navigation task from Gibson (Xia et al., 2018) and iGibson 1.0 (Shen et al., 2021) as well as the DeepMind control suite (DMControl) (Tunyasuvunakool et al., 2020). Our main contributions are:

1. to propose a world model with invariant causal features, which outperforms state-of-the-art models on out-of-distribution generalization and sim-to-real transfer of learned features.

2. to propose intervention invariant auxiliary tasks to improve the performance.

3. to show that world model benefits from contrastive unsupervised representation learning.

## 2 RELATED WORK

**Unsupervised Representation Learning.** Learning reusable feature representations from large unlabeled data has been an active research area. In the context of computer vision, one can leverage unlabeled images and videos to learn good intermediate representations, which can be useful for a wide variety of downstream tasks. A classic approach to unsupervised representation learning is clustering similar data together, for example, using K-means (Hartigan & Wong, 1979). Recently, VAE (Kingma & Welling, 2014) has been a preferred approach for representation learning in model-based RL Ha & Schmidhuber (2018). Since VAE does not make any additional consideration of downstream tasks, invariant representation learning with contrastive loss has shown more promising results (Anand et al., 2019; Laskin et al., 2020a). Further, building on the success of supervised deep learning, getting supervision from the data itself is a preferred approach of representation learning from the unlabelled data, which is known as self-supervised learning. Self-supervised learning formulates the learning as a supervised loss function. In image-based learning self-supervision can be formulated using different data augmentations, for example, image distortion and rotation (Dosovitskiy et al., 2014; Chen et al., 2020). We also use different data augmentation techniques to learn the invariant features using contrastive loss, which we explain below.

**Contrastive Learning.** Representation learning methods based on contrastive loss (Chopra et al., 2005) have recently achieved state-of-the-art performances on face verification task. These methods use a contrastive loss to learn representations invariant to data augmentation (Chen et al., 2020; He et al., 2020). Given a list of input samples, contrastive loss forces samples from the same class to have similar embeddings and different ones for different classes. Since class labels are not available in the unsupervised setting, contrastive loss forces similar embedding for the augmented version of the same sample and different ones for different samples. In a contrastive learning setting, an augmented version of the same sample is known as a positive sample and different samples as a negative sample; these are also referred to as query and key samples, respectively. There are several ways of formulating the contrastive loss such as Siamese (Chopra et al., 2005), InfoNCE (van den Oord et al., 2018) and SimCLR (Chen et al., 2020). We chose InfoNCE (van den Oord et al., 2018) for our contrastive loss in this work.

**Causal Inference using Invariant Prediction.** Learning structured representations that capture the underlying causal mechanisms generating data is a central problem for robust machine learning systems (Schölkopf et al., 2021). However, recovering the underlying causal structure of the environment from observational data without additional assumptions is a complex problem. A recent successful approach for causal discovery, in the context of unknown causal structure, is causal inference using invariant prediction (Peters et al., 2016). The invariance idea is closely linked to causality under the terms *autonomy* and *modularity* (Pearl, 2009) or *stability* (Dawid & Didelez, 2010). Furthermore, it is well known that causal variables have an invariance property, and this group of methods try to exploit this fact for causal inference. Our proposed world model also exploits this fact to learn the causal state of the environment using the invariance principle, which is formalized using contrastive loss. Also, this view of self-supervised representation learning using invariant causal mechanisms is recently formalized by Mitrovic et al. (2021). In section 3, we explain how we utilized the data augmentation technique to learn the causal state of the environment.

**Model-based RL.** The human brain discovers the hidden causes underlying an observation. Those internal representations of the world influence how agents infer which actions will lead to a higher reward (Hamrick, 2019). An early instantiation of this idea was put forward by Sutton (Sutton, 1990), where future hallucination samples rolled out from the learned world model are used in addition to the agent's interactions for sample efficient learning. Recently, planning through the world model has been successfully demonstrated in the *world model* by Ha & Schmidhuber (2018) and *DreamerV2* by Hafner et al. (2021). In our work we propose to learn causal features to reduce the training biases and improve the sample efficiency further.

**Sample Efficiency.** Joint learning of auxiliary tasks with model-free RL makes them competitive with model-based RL in terms of sample efficiency. For example, the recently proposed model-free RL method called CURL (Laskin et al., 2020a) added contrastive loss as an auxiliary task and outperformed the state-of-the-art model-based RL method called Dreamer (Hafner et al., 2020). Also, two recent works using data augmentation for RL called RAD (Laskin et al., 2020b) and DrQ (Yarats et al., 2021) outperform CURL without using an auxiliary contrastive loss. These results

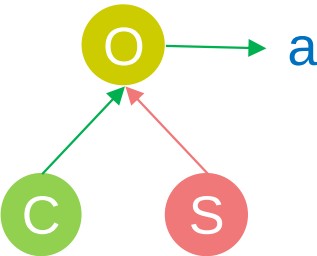

Figure 2: Observation is made of content (C), causal variables, and style (S), spurious variables. We want representation learning to extract the content variables only, *i.e.* true cause of the action (a).

warrant that if an agent has access to a rich stream of data from the environments, an additional auxiliary loss is unnecessary, since directly optimizing the policy objective is better than optimizing multiple objectives. However, we do not have access to a rich stream of data for many interesting problems, hence sample efficiency still matters. Further, these papers do not consider the effect of auxiliary tasks and unsupervised representation learning for model-based RL, which is the main focus of this paper.

## 3 PROPOSED MODEL

The data flow diagram of our proposed world model is shown in Figure 1. We consider the visual control task as a finite-horizon partially observable Markov decision process (POMDP). We denote observation space, action space and time horizon as $\mathcal{O}$, $\mathcal{A}$ and $\mathcal{T}$ respectively. An agent performs continuous actions $a_t \sim p(a_t|o_{\leq t}, a_{<t})$, and receives observations and scalar rewards $o_t, r_t \sim p(o_t, r_t|o_{<t}, a_{<t})$ from the unknown environment. We use the deep neural networks for the state abstractions from observation $s_t \sim p(s_t|s_{t-1}, a_{t-1}, o_t)$, predictive transition model $s_t \sim q(s_t|s_{t-1}, a_{t-1})$ and reward model $r_t \sim q(r_t|s_t)$. The goal of an agent is to maximize the expected total rewards $E_p(\sum_{t=1}^{T} r_t)$. In the following sections we describe our proposed model in detail.

### 3.1 INVARIANT CAUSAL FEATURES LEARNING

Learning the pixel-based controller involves two main tasks, first abstraction of the environment state and second maximizing the expected total reward of the policy. Since visual control is a complex task, the number of causal variables that involve are high, making the causal structure discovery a difficult task as it requires fitting the graphical model or structural equation. That is why we choose invariant prediction for causal feature learning. The key idea is that if we consider all *causes* of an *effect*, then the conditional distribution of the *effect* given the *causes* will not change when we change all the other remaining variables of the system (Peters et al., 2016). Such experimental changes are known as interventions in the causality literature. We have used data augmentation as our mechanism of intervention, since usually we do not have access to the causal or spurious variables or both of the target environment. We have also explored texture randomization as an intervention in our experiments, which we call action replay.

We have shown the high-level idea of the proposed causal features extraction technique in Figure 2. The main idea is observation is made of content (C), causal variables, and style (S), spurious variables. We want our representation learning to extract the content variables only, which are the true cause of the action. In other words we want our control policy to learn $P(a|c = invariant\_encoder(o))$ but not $P(a|(c, s) = encoder(o))$ as causal variables are sample efficient, and robust to OoD generalization and sim-to-real transfer. We have chosen the contrastive learning technique (Chen et al., 2020; Grill et al., 2020) to learn the invariant causal features, which means embedding features of a sample and its data augmented version should be the same. We use spatial jitter (crop and translation), Gaussian blur, color jitter, grayscale and cutout data augmentation techniques (Chen et al., 2020; Laskin et al., 2020b; Yarats et al., 2021) for contrastive learning. Since the intervention of the style is performed at the image level, we do not need to know

the location of the interventions. Further, the data also can come from observation of the different experimental environments. The theoretical guarantee of causal inference using invariant prediction is discussed in Peters et al. (2016) and Mitrovic et al. (2021). However, our proposed method does not consider the hidden confounding variables that influence the target effect variable.

## 3.2 WORLD MODEL

The proposed *World Model with invariant Causal features* (WMC) consists of three main components: i) unsupervised causal representation learning, ii) memory, and iii) controller. The representation learning module uses the contrastive learning for invariant causal state learning. The memory module uses a recurrent neural network. It outputs the parameters of a categorical distribution, which is used to sample the future states of the model. Finally, the controller learns the action probability to maximize the expected reward using an actor critic approach (Williams, 1992; Hafner et al., 2021). We have adopted DreamerV2 (Hafner et al., 2021) to test our proposed hypothesis, which we describe below.

**Unsupervised causal features learning.** Invariant causal feature extraction from the RGB image observation is a key component of our model. As described previously, we learn invariant causal features by maximizing agreement between different style interventions of the same observation via a contrastive loss in the latent feature space. Since the world model optimizes feature learning and controller separately to learn better representative features for downstream tasks, our early experiment with only rewards prediction was poor, which we verified in our experiments. Another reason for separate training of the world model and controller is that most of the complexity resides in the world model (features extraction and memory), so that controller training with RL methods will be easier. Hence, we need a stronger loss function to learn a good state representation of the environment. That is why we propose depth reconstruction as an auxiliary task and do not use image resize data augmentation to keep the relation of object size and distance intact. We used InfoNCE (van den Oord et al., 2018) style loss to learn the invariant causal feature. Hence, our encoder takes RGB observation and task specific information as inputs and depth reconstruction and reward prediction as targets. The invariant state abstraction is enforced by contrastive loss. The proposed invariant causal features learning technique has the following three major components:

- A *style intervention* module that uses data augmentation techniques. We use spatial jitter, Gaussian blur, color jitter, grayscale and cutout data augmentation techniques for style intervention. Spatial jitter is implemented by first padding and then performing random crop. Given any observation $o_t$, our style intervention module randomly transforms it into two correlated views of the same observations. All the hyperparameters are provided in the appendix.

- We use an *encoder* network that extracts representations from augmented observations. We follow the same configurations as DreamerV2 for a fair comparison, and only contrastive loss and depth reconstruction tasks are added. We obtain $\tilde{s}_t = invariant\_encoder(o_t)$, then we follow the DreamerV2 to obtain the final state $s_t \sim p(s_t|s_{t-1}, a_{t-1}, \tilde{s}_t)$. We use the contrastive loss immediately after the encoder $\tilde{s}_t$.

- *Contrastive loss* is defined for a contrastive prediction task, which can be explained as a differentiable dictionary lookup task. Given a query observation $q$ and a set $K = \{k_0, k_1, ...k_{2B}\}$ with known positive $\{k_+\}$ and negative $\{k_-\}$ keys, the aim of contrastive loss is to learn a representation in which positive sample pairs stay close to each other while negative ones are far apart. In contrastive learning literature $q$, $K$, $k_+$ and $k_-$ are also referred as *anchors*, *targets*, *positive* and *negative* samples. We use bilinear products for *projection head* and InfoNCE loss for contrastive learning (van den Oord et al., 2018), which enforces the desired similarity in the embedding space:

$$\ell_t^q = \log \frac{\exp(q^T W k_+)}{\exp(q^T W k_+) + \sum_{i=0}^{2(B-1)} \exp(q^T W k_i)} \tag{1}$$

This loss function can be seen as the log-loss of a $2B$-way softmax classifier whose label is $k_+$. Where $B$ is a batch size, which becomes $2B$ after the style intervention module randomly transforms in two correlated views of the same observation. The quality of features using contrastive loss depends on the quality of the negative sample mining, which is a difficult task in an unsupervised setting. We slice the sample observations from an episode at

each 5th time-step to reduce the similarity between neighbouring negative observations. In summary, causal feature learning has the following component and is optimized by Equation 1.

$$\text{Invariant causal model:} \quad p_\theta(\tilde{s}_t|o_t) \tag{2}$$

**Future predictive memory model.** The representation learning model extracts what the agent sees at each time frame but we also want our agent to remember the important events from the past. This is achieved with the memory model and implemented with a recurrent neural network. Further, the transition model learns to predict the future state using the current state and action in the latent space only, which enables future imagination without knowing the future observation since we can obtain the future action from the policy if we know the future state. Hence, this module is called a future predictive model and enables efficient latent imagination for planning (Ha & Schmidhuber, 2018; Hafner et al., 2020). In summary, dynamic memory and representation learning modules are tightly integrated and have the following components,

$$
\begin{aligned}
\text{Representation model:} & \quad p_\theta(s_t|s_{t-1}, a_{t-1}, \tilde{s}_t) \\
\text{Depth prediction model:} & \quad q_\theta(o_t^d|s_t) \\
\text{Reward model:} & \quad q_\theta(r_t|s_t) \\
\text{Predictive memory model:} & \quad q_\theta(s_t|s_{t-1}, a_{t-1}).
\end{aligned} \tag{3}
$$

All the world model and representation losses were optimized jointly, which includes contrastive, depth prediction, reward and future predictive KL regularizer losses respectively,

$$\mathcal{L}_{WM} = E_p\left(\sum_t \left(\ell_t^q + \ln q(o_t^d|s_t) + \ln q(r_t|s_t) - \beta\ell_t^{KL}\right)\right) \tag{4}$$

where, $\ell_t^{KL} = KL(p(s_t|s_{t-1}, a_{t-1}, \tilde{s}_t)||q(s_t|s_{t-1}, a_{t-1}))$

**Controller.** The objective of the controller is to optimize the expected rewards of the action, which is optimized using an actor critic approach. The actor critic approach considers the rewards beyond the horizon. Since we follow the DreamerV2, an action model and a value model are learnt in the imagined latent space of the world model. The action model implements a policy that aims to predict future actions that maximizes the total expected rewards in the imagined environment. Given $H$ as the imagination horizon length, $\gamma$ the discount factor for the future rewards, action and policy model are defined as follows:

$$
\begin{aligned}
\text{Action model:} & \quad q_\phi(a_t|s_t) \\
\text{Value model:} & \quad E_{q(\cdot|s_\tau)}\sum_{\tau=t}^{t+H} \gamma^{\tau-t} r_\tau.
\end{aligned} \tag{5}
$$

### 3.3 IMPLEMENTATION DETAILS

We have used the publicly available code of DreamerV2 (Hafner et al., 2021) and added the contrastive loss on top of that. We have used default hyperparameters of the continuous control task. Here, we have explained the necessary changes for our experiments. Following MoCo (He et al., 2020) and BYOL (Grill et al., 2020) we have used the moving average version of the query encoder to encode the keys $K$ with a momentum value of 0.999. The contrastive loss is jointly optimized with the world model using Adam (Kingma & Ba, 2014). To encode the task observations we used two dense layers of size 32 with ELU activations (Clevert et al., 2015). The features from RGB image observation and task observation are concatenated before sending to the representation module of the DreamerV2. Replay buffer capacity is $3e^5$ for both 100k and 500k steps experiments. All architectural details and hyperparameters are provided in the appendix. Further, the training time of WMC is almost twice than that of DreamerV2 but inference time is the same.

Table 1: Experiment results on PointGoal navigation task from iGibson 1.0 dataset. Even though data augmentation (DA) improves the DreamerV2 with depth (D) reconstruction rather than RGB image(I), proposed WMC further improves the results by a significant margin.

| Models | Steps | Ihlen_0_int | | Ihlen_1_int | | Rs_int | | Env Avg | |
|---|---|---|---|---|---|---|---|---|---|
| | | SR | SPL | SR | SPL | SR | SPL | SR | SPL |
| RAD | 100k | 0.5 | 0.01 | 0.1 | 0.00 | 0.8 | 0.01 | 0.5 | 0.01 |
| CURL | 100k | 8.0 | 0.07 | 0.6 | 0.00 | 5.4 | 0.05 | 4.6 | 0.04 |
| DreamerV2 | 100k | 1.7 | 0.01 | 0.5 | 0.00 | 1.6 | 0.01 | 1.3 | 0.01 |
| DreamerV2 + DA | 100k | 7.2 | 0.05 | 1.5 | 0.01 | 7.7 | 0.05 | 5.5 | 0.03 |
| DreamerV2 - I + D | 100k | 1.9 | 0.01 | 0.9 | 0.00 | 2.8 | 0.01 | 1.9 | 0.01 |
| DreamerV2 - I + D + DA | 100k | 8.3 | 0.05 | 2.1 | 0.01 | 10.8 | 0.07 | 7.0 | 0.04 |
| WMC | 100k | **28.9** | **0.22** | **7.9** | **0.05** | **30.2** | **0.22** | **22.3** | **0.16** |
| RAD | 500k | 48.7 | **0.44** | 11.6 | **0.11** | 48.5 | 0.44 | 36.3 | 0.32 |
| CURL | 500k | 40.8 | 0.36 | 11.4 | 0.09 | 41.9 | 0.36 | 31.4 | 0.27 |
| DreamerV2 | 500k | 1.3 | 0.01 | 0.8 | 0.00 | 2.2 | 0.01 | 1.4 | 0.01 |
| DreamerV2 + DA | 500k | 7.2 | 0.04 | 1.1 | 0.01 | 9.7 | 0.05 | 13.0 | 0.08 |
| DreamerV2 - I + D | 500k | 3.3 | 0.02 | 0.9 | 0.00 | 4.1 | 0.02 | 2.7 | 0.01 |
| DreamerV2 - I + D + DA | 500k | 38.0 | 0.25 | 9.0 | 0.06 | 52.7 | 0.35 | 33.2 | 0.22 |
| WMC | 500k | **58.6** | **0.44** | **15.7** | **0.11** | **67.4** | **0.51** | **47.2** | **0.36** |

Table 2: iGibson-to-Gibson dataset: sim-to-real perception transfer results on navigation task.

| Models | Steps | Ihlen | | Muleshoe | | Uvalda | | Noxapater | | McDade | |
|---|---|---|---|---|---|---|---|---|---|---|---|
| | | SR | SPL | SR | SPL | SR | SPL | SR | SPL | SR | SPL |
| RAD | 100k | 0.0 | 0.00 | 0.0 | 0.00 | 0.0 | 0.00 | 0.0 | 0.00 | 0.0 | 0.00 |
| CURL | 100k | 5.9 | 0.05 | 3.8 | 0.03 | 5.1 | 0.04 | 5.9 | 0.05 | 12.8 | 0.11 |
| WMC | 100k | **24.4** | **0.18** | **20.4** | **0.15** | **24.3** | **0.18** | **27.3** | **0.21** | **40.9** | **0.31** |
| RAD | 500k | 26.4 | 0.23 | 27.5 | 0.24 | 28.5 | 0.25 | 28.6 | 0.25 | 40.0 | 0.34 |
| CURL | 500k | 36.8 | 0.33 | 29.3 | 0.27 | 33.7 | 0.30 | 35.2 | 0.32 | 53.8 | 0.50 |
| WMC | 500k | **50.0** | **0.38** | **50.3** | **0.38** | **49.7** | **0.37** | **45.5** | **0.34** | **50.7** | **0.38** |

# 4 EXPERIMENTS

## 4.1 EVALUATION

We evaluate the out-of-distribution (OoD) generalization, sim-to-real transfer of perception learning and sample-efficiency of our model and baselines at 100k and 500k environment steps. Sample efficiency test on 100k and 500k steps is a common practice (Laskin et al., 2020a;b; Yarats et al., 2021; Hafner et al., 2021). Following (Hafner et al., 2021), we update the model parameters on every fifth interactive step. We used default hyperparameter values of DreamerV2 for our experiments. Similarly, we used official code for RAD and CURL experiments.

## 4.2 IGIBSON DATASET

We have tested our proposed WMC on a random *PointGoal* task from iGibson 1.0 environment (Shen et al., 2021) for OoD generalization. It contains 15 floor scenes with 108 rooms. The scenes are replicas of real-world homes with artist designed textures and materials. We have used RGB, depth and task related observation only. Depth is only used during the training phase. The task related observation includes goal location, current location, and linear and angular velocities of the robot. Action includes rotation in radians and forward distance in meters for the Turtlebot. Since iGibson 1.0 does not provide dataset splits for OoD generalization, we have chosen five scenes for

Table 3: Experiment results on DMControl. Results are reported as averages across 10 seeds.

| 100k Steps Total Rewards | WMC | CURL | Dreamer | SAC+AE | PSAC | SSAC |
|---|---|---|---|---|---|---|
| Finger, spin | 486±191 | **767±56** | 341±70 | 740±64 | 179±66 | 811±46 |
| Cartpole, swingup | 472±67 | **582±146** | 326±27 | 311±11 | 419±40 | 835±22 |
| Reacher, easy | 327±98 | **538±233** | 314±155 | 274±14 | 145±30 | 746±25 |
| Cheetah, run | **321±78** | 299 ±48 | 235± 137 | 267±24 | 197±15 | 616±18 |
| Walker, walk | **654±100** | 403±24 | 277±12 | 394±22 | 42±12 | 891±82 |
| Ball in cup, catch | **830±118** | 769±43 | 246±174 | 391±82 | 312±63 | 746±91 |
| 500K Steps Total Rewards | | | | | | |
| Finger, spin | 471±173 | **926±45** | 796±183 | 884±128 | 179±166 | 923±21 |
| Cartpole, swingup | 675±64 | **841±45** | 762±27 | 735±63 | 419±40 | 848±15 |
| Reacher, easy | 891±72 | **929±44** | 793±164 | 627±58 | 145±30 | 923±24 |
| Cheetah, run | **633±70** | 518±28 | 570±253 | 550±34 | 197±15 | 795±30 |
| Walker, walk | **965±4** | 902±43 | 897±49 | 847±48 | 42±12 | 948±54 |
| Ball in cup, catch | 950±20 | **959±27** | 879± 87 | 794± 58 | 312± 63 | 974±33 |

Table 4: Ablation study of the proposed WMC, the results clearly show the benefit of intervention-invariant auxiliary task, depth D, and action replay (AR).

| Models | Steps | Ihlen_0_int | | Ihlen_1_int | | Rs_int | | **Env Avg** | |
|---|---|---|---|---|---|---|---|---|---|
| | | SR | SPL | SR | SPL | SR | SPL | SR | SPL |
| WMC | 100k | 28.9 | 0.22 | 7.9 | 0.05 | 30.2 | 0.22 | 22.3 | 0.16 |
| WMC - AR | 100k | 15.4 | 0.11 | 4.2 | 0.02 | 19.7 | 0.12 | 13.1 | 0.08 |
| WMC - AR - D | 100k | 0.0 | 0.0 | 0.0 | 0.0 | 0.0 | 0.0 | 0.03 | 0.00 |
| WMC - D + I | 100k | 15.4 | 0.11 | 4.6 | 0.03 | 17.0 | 0.12 | 12.3 | 0.09 |
| WMC | 500k | 58.6 | 0.44 | 15.7 | 0.11 | 67.4 | 0.51 | 47.2 | 0.36 |
| WMC - AR | 500k | 45.1 | 0.28 | 11.5 | 0.07 | 26.8 | 0.18 | 27.8 | 0.17 |
| WMC - AR - D | 500k | 0.9 | 0.0 | 0.2 | 0 | 1.2 | 0.01 | 0.7 | 0.00 |
| WMC - D + I | 500k | 28.6 | 0.12 | 6.6 | 0.04 | 22.3 | 0.14 | 19.1 | 0.10 |

training and tested on the held-out three scenes and visual textures both. The details are provided in the appendix.

We have trained all models three times with random seeds and report the average *Success Rate* (SR) and *Success weighted by (normalized inverse) Path Length* (SPL) on held-out scenes as well as visual textures in the Table 1. Our proposed WMC outperforms state-of-the-art model-based RL method DreamerV2 and model-free method RAD and CURL on 100k and 500k interactive steps. Even though depth reconstruction and data augmentation improve the DreamerV2, proposed invariant causal features learning with contrastive loss further improves the results. All methods perform poorly on the difficult Ihlen_1_int scene. Further, our experimental results confirm that, similar to the model-free RL methods (Laskin et al., 2020b), data augmentation improves the performance of the model-based RL.

## 4.3 IGIBSON-TO-GIBSON DATASET

We use the Gibson dataset (Xia et al., 2018) for sim-to-real transfer experiments of the perception module, representation learning module of the world model; however please note that the robot controller is still a part of the simulator. Gibson scenes are created by 3D scanning of the real scenes, and it uses a neural network to fill the pathological geometric and occlusion errors only. We have trained all models on the artist created textures of iGibson and tested on five scenes from the Gibson. The results are shown in Table 2. Our proposed WMC outperforms RAD and CURL on

100k and 500k interactive steps, which shows that WMC learns more stable features and is better suited for sim-to-real transfer.

## 4.4 DEEPMIND CONTROL SUITE

The results for the DMControl suite (Tunyasuvunakool et al., 2020) experiments are shown in Table 3. We replaced the depth reconstruction of WMC with the original RGB input reconstruction to adopt the DMControl suite. WMC achieved competitive results, and the key findings are: i) even though depth reconstruction is an important component to enforce the invariant causal features learning on WMC explicitly, the competitive results, even without depth reconstruction show the wider applicability of the proposed model; ii) WMC is competitive with CURL, the closest state-of-the-art RL method with contrastive learning.

## 4.5 ABLATION STUDY

In many real world environments, action replay with texture randomization is not possible to perform. Hence, we have also experimented WMC without action replay (AR) feature. The results are shown in Table 4. These experiments confirm that WMC learning with only standard self-supervised learning is better than just data augmentation. However, in 500k steps measure data augmentation technique on DreamerV2 with depth reconstruction is doing better in Rs_int environment. This result suggests that model optimization with an additional constraint makes the optimization task harder. However, collecting more interactive data in the real environment is difficult in many scenarios. Since given the simulation to real gap is reducing every year (Shen et al., 2021) and WMC is performing better in 5 out of 6 results, an additional contrastive loss is still important for the model-based RL. Further, this result warrants that we need a better method for intervention on image style or spurious variables. Since the separation of those variables from the pixel-observation is not an easy task, we do need some new contributions here, which we leave as future work.

The standard formulation of contrastive learning does not use reconstruction loss (Chen et al., 2020; Grill et al., 2020; Laskin et al., 2020a; He et al., 2020). Since model-based RL does not optimize the representation learning and controller jointly, contrastive loss collapses. Hence, to validate our proposal of style intervention at observation space and intervention invariant depth reconstruction as an auxiliary task, we have done experiments without depth reconstruction and action replay (WMC - AR - D). The results are shown in Table 4. The degradation in performance without depth reconstruction loss is much worse than only without action replay. In summary, in a reasonably complex pixel-based control task, WMC is not able to learn meaningful control. This shows the value of our careful design of the WMC using contrastive loss and depth reconstruction as an auxiliary task.

Further, WMC with RGB image reconstruction rather than depth (WMC - D + I) reduces the success rate by 50% and SPL by almost two third. These results confirm that our proposal of doing an intervention on RGB observation space and adding intervention invariant reconstruction of depth as an auxiliary task is one of the key contributions of our paper.

## 5 CONCLUSION

In this work we proposed a method to learn *World Model with invariant Causal features* (WMC). These invariant causal features are learnt by minimizing contrastive loss between content invariance interventions of the observation. Since the world model learns the representation learning and policy of the agent independently, without providing the better supervisory signal for the representation learning module, the contrastive loss collapses. Hence, we proposed depth reconstruction as an auxiliary task, which is invariant to the proposed data augmentation techniques. Further, given an intervention in the observation space, WMC can *extract* as well as *predict* the related causal features.

Our proposed WMC significantly outperforms the state-of-the-art models on out-of-distribution generalization, sim-to-real transfer of perception module and sample efficiency measures. Further, our method works on a sample-level intervention and does not need data from different environments to learn the invariant causal features.

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

# A APPENDIX

## A.1 QUALITATIVE RESULTS OF WMC

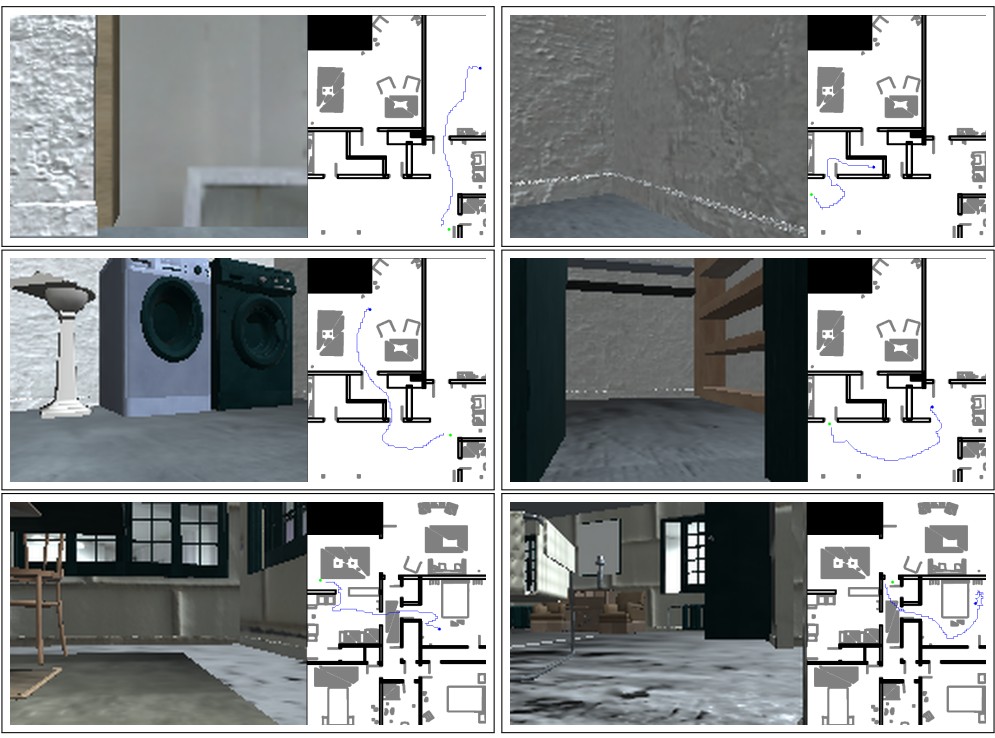

Figure 3: The out-of-distribution generalization tests of proposed WMC on held-out scenes and visual textures from iGibson 1.0 environment. Green circle is a random *PointGoal*, blue circle is a random starting point and blue line represents the travel path of the Turtlebot robot.

## A.2 HYPER PARAMETERS

Table 5: Hyper parameters of proposed WMC.

| Name | Symbol | Value |
|---|---|---|
| **World Model** | | |
| Dataset size (FIFO) | — | $3 \cdot 10^5$ |
| iGibson input image size | $o$ | $120{\times}160$ |
| Batch size | $B$ | 50 |
| Sequence length | $L$ | 50 |
| Discrete latent dimensions | — | 32 |
| Discrete latent classes | — | 32 |
| RSSM number of units | — | 1024 |
| KL loss scale | $\beta$ | 1.0 |
| World model learning rate | — | $3 \cdot 10^{-4}$ |
| Key encoder exponential moving average | — | 0.999 |
| **Behavior** | | |
| Imagination horizon | $H$ | 15 |
| Actor learning rate | — | $1 \cdot 10^{-4}$ |
| Critic learning rate | — | $1 \cdot 10^{-4}$ |
| Slow critic update interval | — | 100 |
| **Common** | | |
| Policy steps per gradient step | — | 4 |
| Policy and reward MPL number of layers | — | 4 |
| Policy and reward MPL number of units | — | 400 |
| Gradient clipping | — | 100 |
| Adam epsilon | $\epsilon$ | $10^{-5}$ |
| **Encoder and Decoder** | | |
| MLP encoder sizes of task obs | — | 32, 32 |
| Encoder kernels sizes | — | 4, 4, 4, 4, 4 |
| Decoder kernels sizes | — | 5, 5, 4, 5, 4 |
| Encoder and decoder feature maps | — | 32, 64, 128, 256, 512 |
| Encoder and decoder strides | — | 2, 2, 2, 2, 2 |
| Decoder padding | — | none, 0-1, none, none, none |
| **Data Augmentation** | | |
| Padding range | — | 10 |
| Hue delta | — | 0.1 |
| Brightness delta | — | 0.4 |
| Contrast delta | — | 0.4 |
| Saturation delta | — | 0.2 |
| Gaussina blur sigma min, max | — | 0.1, 2.0 |
| Cutout min, max | — | 30, 50 |

## A.3 iGibson 1.0 Training and Evaluation Splits

Table 6: Train-test scenes splits for iGibsion 1.0 dataset (Shen et al., 2021).

| Phase | Scene names |
|---|---|
| Training | Beechwood_0_int, Beechwood_1_int, |
| | Benevolence_0_int, Benevolence_1_int, Benevolence_2_int |
| | Merom_0_int, Merom_1_int, |
| | Pomaria_0_int, Pomaria_1_int, Pomaria_2_int, |
| | Wainscott_0_int, Wainscott_1_int |
| Testing | Ihlen_0_int, Ihlen_1_int, Rs_int |

Table 7: iGibson 1.0 environment (Shen et al., 2021) held out texture ids for test.

| Material category | Held-out texture ids for test |
|---|---|
| asphalt | 06, 15 |
| bricks | 08, 19 |
| concrete | 06, 15, 17 |
| fabric | 01, 02, 28 |
| fabric_carpet | 02, 05, 13 |
| ground | 13, 19 |
| leather | 03, 12 |
| marble | 02, 03 |
| metal | 10, 19 |
| metal_diamond_plate | 04 |
| moss | 01, 03 |
| paint | 05 |
| paving_stones | 24, 38 |
| planks | 07, 09, 16 |
| plaster | 03 |
| plastic | 04, 05 |
| porcelain | 02, 04 |
| rocks | 04 |
| terrazzo | 06, 08 |
| tiles | 43, 49 |
| wood | 02, 05, 16, 22, 32 |
| wood_floor | 06, 10, 17, 28 |

Table 8: Examples of textures split for training and testing from the proposed split in Table 7.

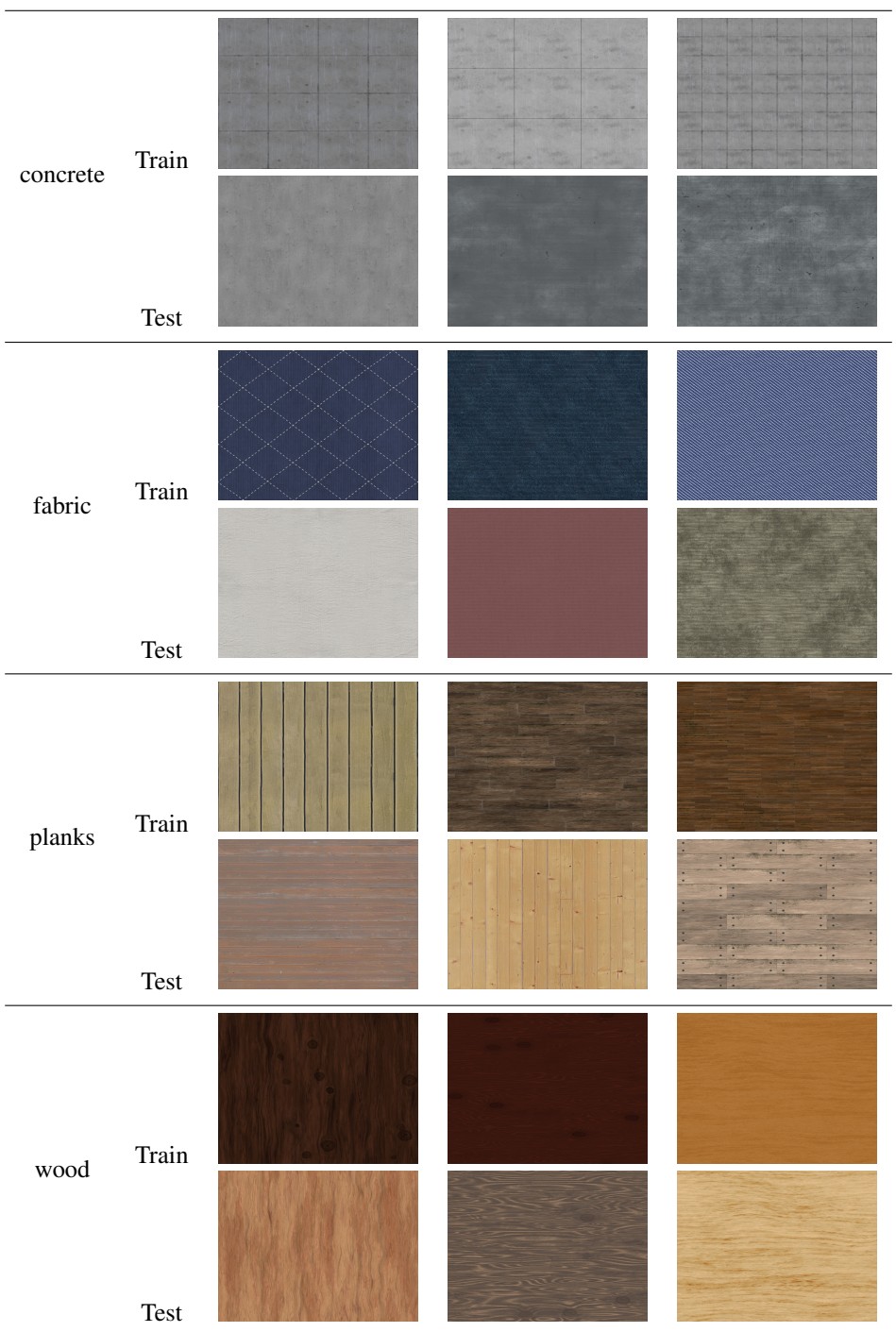

