# OpenReview forum: "Contrastive Unsupervised Learning of World Model with Invariant Causal Features"
_ICLR.cc/2023/Conference — Submitted to ICLR 2023_

### Official Review · Reviewer_Hdwq · 2022-10-16

**Confidence:** 5
**Correctness:** 2
**Technical Novelty And Significance:** 2
**Empirical Novelty And Significance:** 2
**Recommendation:** 3

**Clarity, Quality, Novelty And Reproducibility:**

- Clarity: see above
- Quality: Although the findings are impressive (perhaps not surprising), I believe the idea lacks originality and the language is poor.
- Novelty: see above
- Reproducibility: As far as I can see, all experiment details are included. The implementation is also based on published DreamerV2 code. So, the results should be reproducible.

**Strength And Weaknesses:**

Strengths:
- The results tables are impressive. Given a fixed compute budget (e.g., environment steps), the method outperforms SOTA or performs on par.

Weaknesses:
- Lack of originality: The method is very largely built upon Dreamer - a SOTA MBRL method. The major improvements over Dreamer are (i) the contrastive objective and (ii) the depth prediction task. The former seems to be a very straightforward application of Mitrovic'21 - so much so that Mitrovic'21 presents a very similar version of Figure-2. I also believe the depth prediction task is a rather practical aspect of the work, which significantly restricts the dataset characteristics. As such, the overall contribution is a combination of well-known optimization targets, which I believe lacks novelty in order to be published in this venue.
- The writing should be significantly improved. Both grammar and text structure requires modifications. Here are some tips:
  - Contribution and motivation are mixed up in the abstract. The first two sentences state the contribution, then the motivation, and then we read the contribution again.
  - "Specifically, we utilize depth prediction to explicitly enforce the invariance and use data augmentation as style intervention on the RGB observation space." Are these two parts of the sentence related? Otherwise I suggest having two different sentences.
  - "Our design leverages unsupervised representation learning to learn the world model with invariant causal features" was already stated in the abstract.
  - The first paragraph is a discussion on model-free vs model-based RL, which is not directly related to the rest of the paper.
  - "... as the control policy is learned end-to-end." What does end-to-end refer to here? Perhaps learning from pixels?
  - "suited to separate relevant states from irrelevant ones" Relevant to what? Policy learning?
  - "so inference of the underlining causal structure" Underlying, perhaps?
  - Short explanations for the components in Figure-1 would be nice to include in the caption.
  - "Zhang et al. (2020) uses multiple environments to learn the invariant causal features using a common encoder. Here spurious or irrelevant features are learnt using environment specific encoders." This reads like related work instead of intro, making it more difficult to follow this long paragraph.
  - "Model-based RL methods do not optimize feature learning" This is vague: What does it mean to "optimize feature learning"?
  - "The aim is that such features will be" Which features?
  - "Hence, we propose a causally invariant auxiliary task for invariant causal features learning." What is the connection with the previous part (why hence?)
  - "Further, most computer vision data augmentations on RGB image space are invariant to depth." Why is this important?
  - "Our main contributions are to show ... to propose ..."
  - "Representation learning methods based on contrastive loss (Chopra et al., 2005) have recently achieved state-of-the-art performances" on what tasks?
  - "Learning structured representations that" capture instead of captures
  - "Furthermore, it is well known that causal variables have an invariance property" This requires elaboration as it might not be familiar to some readers.
  - "Sample Efficiency" should be in bold.
  - "model-based RL **method** called"
  - "Further, these papers do not consider the effect of data augmentation" whereas both RAD and CURL utilize augmentations?
  - What is "a" in Figure 2?
  - "the number of causal variables **that involve**"
  - "That is why we choose invariant prediction as our method of choice for causal feature learning." Perhaps say "That is why we choose invariant prediction for causal feature learning"?
  - "Such experimental changes are known as interventions" Isn't it a bit late to describe interventions? Maybe explain it in the intro?
  - "since we do not have access to the causal or spurious variables or both of the environment." Which environments?
  - "..., which we call action replay." Why this name?
  - "The main idea is observation is made of content (C), causal variables, and style (S), spurious variables" This is unclear: aren't C and S latent factors?
  - "which means embedding features of a sample and its data augmented version, intervention on style or spurious variables, should be the same" This is unclear.
  - "the data also can come from observation of the different environments" Unclear, which environments?
  - "with different environment-level interventions" Why would that be the case? Would environment-specific interventions be a by-product of having multiple environments or because of our interventions?
  - "memory also know as world model" Unclear, why is memory known as world model?
  - "Memory module uses **a** recurrent neural network"
  - "parameters of a categorical distribution, discrete and multinomial distribution," Discrete distribution?
  - "The controller maximizes the action probability" Perhaps learns over maximizes?
  - "our early result with only rewards prediction was poor" What early results?
  - "and reward prediction as target." Perhaps "rewards as predicted targets"?
  - The list of used data augmentations is mentioned twice.
  - "transforms in two correlated views"
  - "that extracts representation **s** from"
  - "We use the contrastive loss immediately after the encoder" How do you use it?
  - "Given a query observation q and a set K = {k0, k1, ...k2B} with known positive {k+} and negative {k−} keys." The sentence lacks a verb.
  - "Where B is a batch size, which becomes..."



- Reference is needed to support many claims in the paper:
  -  An important reference to one of the first model-based RL papers is missing: Deisenroth, Marc, and Carl E. Rasmussen. "PILCO: A model-based and data-efficient approach to policy search." Proceedings of the 28th International Conference on machine learning (ICML-11). 2011.
  -  "model-free methods learn a greedy state representation" One would argue that model-free approaches do not even learn a state representation, e.g., if trained on noisy state observations. Giving a reference to support this claim would be better.
  - "Therefore, we consider model-based methods more attractive for continuous learning, out-of-distribution (OoD) generalization and sim-to-real transfer" Reference needed to support the claim.
  - "resulting in performance degradation of the control policy, even for slight task-irrelevant style changes."
  - "so inference of the underlining causal structure of the model becomes intractable."
  - "Recently, VAE (Kingma & Welling, 2014) has been a preferred approach for representation learning in model-based RL"
  - Almost the whole MBRL literature is left out of "Model-based RL" paragraph. I suggest authors explain their scope here.



**Summary Of The Paper:**

Thıs paper presents an approach for learning world models with causal features. The authors propose to augment well-established model-based reinforcement learning (MBRL) objectives with a contrastive unsupervised learning objective, which induces causal features and invariances. Unlike existing approaches, the model is expected to predict depth as a side task, which is shown to improve the model performance. The model outperforms the SOTA MBRL method (Dreamer on out-of-distribution navigation tasks. It is also shown to achieve best results on a sim-to-real navigation task.

**Summary Of The Review:**

Overall, the idea of improving MBRL with representation learning techniques is a nice idea. Authors opt for perhaps the simplest approach (augmenting the MBRL objective with a contrastive learning loss), which restricts the originality of the work. I also believe the writing requires major updates, especially the grammar and (lacking) references.

---

> ### Author Response · Authors · 2022-11-15
> **Addressing your concerns and summary of changes**
>
> Thank you for your comments and pointing out many typos and grammar issues. Below, we answered all of your concerns. We hope this gives you the confidence to support our paper with a higher score.
>
> **Q1** Lack of originality: The method is very largely built upon Dreamer - a SOTA MBRL method. The major improvements over Dreamer are (i) the contrastive objective and (ii) the depth prediction task. The former seems to be a very straightforward application of Mitrovic'21 - so much so that Mitrovic'21 presents a very similar version of Figure-2. I also believe the depth prediction task is a rather practical aspect of the work, which significantly restricts the dataset characteristics. As such, the overall contribution is a combination of well-known optimization targets, which I believe lacks novelty in order to be published in this venue.
> > **A1** Our key insight is `designing intervention invariant auxiliary task` to learn the invariant causal features, which is robust against any changes to the nuisance variables. Hence, reconstruction of depth or denoising RGB input from the various data augmentations (so it's not exactly the input reconstruction) are just two realizations/implementations to verify our proposal. Other examples of intervention invariant auxiliary task can be dense semantic segmentation, dense scene flow and sparse landmarks detection. Hence, the main difference with Mitrovic'21 is that our WMC explicitly forces the invariant causal features learning using intervention invariant auxiliary task. We hope ultimately this idea can be a catalyst to invent the extraction of the relevant variables or causal variables only from the whole observation by designing better intervention invariant auxiliary tasks rather than simply viewing it as a depth or RGB reconstructions/decoders.
>
> **Q2** The writing should be significantly improved. Both grammar and text structure requires modifications.
> > **A2** Thank you for pointing those out. We have update the new version of the paper now, accommodating all of your suggestions. And, we promise to do a full review of text and grammar for the final version without changing the meaning/core of the paper.
>
> **Q3** Reference is needed to support many claims in the paper
> > **A3** Thank you once again, we have updated the paper now with these suggestions.
>
> Thank you so much for appreciating the overall idea of improving MBRL with representation learning techniques. As you have noted perhaps we choose the simplest approach to achieve that goal. In fact, we argue that getting significantly better results with the simplest approach is the strongest part of our model. Further, we believe that contrary to the RAD and DrQ findings, we have shown that MBRL with invariant causal features are more suitable for downstream tasks including sim-to-real than simply more data augmentation. Therefore, we think that especially our sim-to-real results are worth sharing to the community.

---

### Official Review · Reviewer_acab · 2022-10-23

**Confidence:** 3
**Correctness:** 2
**Technical Novelty And Significance:** 1
**Empirical Novelty And Significance:** 2
**Recommendation:** 1

**Clarity, Quality, Novelty And Reproducibility:**

The proposed approach is not very original; it makes two minor modifications to an existing technique. The choice of depth prediction instead of image is not well motivated. And using contrastive learning in an RL setup is not novel. The paper is not very clearly written unfortunately, and some details of the empirical evaluation is not clear enough and makes it difficult to judge some claims of the paper.


**Strength And Weaknesses:**

Unfortunately, I can't find many strengths of this paper. It makes minor modifications to DreamerV2 and shows that these help in a very particular environment.

First, the paper doesn't provide a good motivation for removing image prediction and using depth prediction instead. Given that depth is not generally available, this limits the applicability of the approach significantly. If the argument is that depth prediction in general is better than image prediction, then the evaluations should have made that case (by evaluating model trained with depth in more environments and different settings).

Adding a contrastive learning objective that makes use of data augmentations is not really novel. There are many papers that does this and show it helps. It is unfortunately not clear what the novelty of this paper is wrt to contrastive learning in RL.

The quantitative results on iGibson look promising but it is unclear whether the other contrastive learning techniques (like RAD or CURL) use the same data augmentations. If not, this wouldn't be a fair comparison. Relatedly, is texture randomization used here? If so, is it used by all contrastive learning based techniques?

Unfortunately, the writing needs to be improved as well. Some of the text was difficult to understand (some examples below), and some of the arguments didn't seem to follow. For example, the authors point out (in a couple of places) that their technique learns invariant causal features of the environment. However, this really depends crucially on what interventions are available; if the interventions you have do not cover all of the spurious variables then contrastive learning over data augmentations cannot learn causal features (it will learn a combination of causal and spurious). And in fact there is really nothing special about the technique that allows it to learn these causal features; it is the data augmentations (interventions), which are provided by us (the humans) that allows learning these causal features.

Other points:

- Do the augmentations change all the spurious variables?
- The authors texture randomization "action replay". This seemed rather confusing to me. How is action replay and texture randomization the same thing?
- Do other techniques use the same augmentations? Especially texture randomization?
- For sim-to-real results, do all techniques use texture randomization? Otherwise the comparison is not fair.
- It'd be nice to have std devs with quantitative results.
- Please improve table captions. For example, table 4 caption should mention what abbreviations mean?
- There are other self-supervised approaches for RL slike SpR (Data-Efficient Reinforcement Learning with Self-Predictive Representations). It'd be nice to compare to them.

Typos etc.
pg 3. Self-supervised learning formulates the learning as a supervised loss function. Confusing sentence. Perhaps "as the optimization of a supervised loss function" maybe?
pg 4. variables involves are high -> involved
pg 5. also know as -> known
pg 6. which is optimizes -> optimized
pg 6. The actor critic approach considers the rewards beyond the horizons. Confusing sentence. Many RL techniques consider rewards beyond the horizon.


**Summary Of The Paper:**

This paper adds depth prediction (instead of RGB image prediction) and a contrastive learning objective (with data augmentations) to the world model of DreamerV2 and shows that this results in better performance on the iGibson dataset. For contrastive learning, they use InfoNCE objective and create positive pairs using data augmentations (like crop, blur etc.). One important type of augmentation here is texture randomization. They also remove the image reconstruction loss in DreamerV2 world model training objective and add depth prediction instead. Note this requires depth information to be available in training environment. They compare their technique to plain DreamerV2 and other constrastive learning based methods for RL such as RAD and CURL. They show their technique performs better on iGibson dataset and can transfer better to real textures when trained on synthetic ones (one variant of sim-to-real problem).


**Summary Of The Review:**

Overall the novelty is pretty limited and there are concerns regarding the empirical evaluation, so I don't think the paper is ready for acceptance in its current form unfortunately.

---

> ### Author Response · Authors · 2022-11-15
> **Addressing your concerns and summary of changes (1 of 2)**
>
> Thank you, we have answered all of your concerns below, and we hope this gives you the confidence to support our paper with a higher score.
>
> **Q1** The paper doesn't provide a good motivation for removing image prediction and using depth prediction instead. Given that depth is not generally available, this limits the applicability of the approach significantly.
> > **A1** In this work, we aim to learn invariant causal features. The invariance is forced in two ways: i) implicitly using contrastive loss, and ii) explicitly using intervention (data augmentation) invariant auxiliary tasks. Since, depth prediction is invariant to most of the computer vision data augmentation techniques and available in most of the robotics simulators, we proposed depth prediction as an auxiliary task. However, please note that instead of depth we can also use any other intervention invariant auxiliary task, for example semantic segmentation or scene flow depending on tasks and available data. We have also shown in the DeepMind control suite experiments, where depth is not available, we can use image denoising as an auxiliary task. (Note, since the original image is to be reconstructed from augmented image input, it still finds intervention invariant features).
>
> **Q2** It is unfortunately not clear what the novelty of this paper is wrt to contrastive learning in RL.
> > **A2** Further to answer #A1, we use contrastive learning to "implicitly" force the invariace. However, our main contribution is proposing intervention invariant auxiliary tasks to "explicitly" force the invariance. We propose these two approaches to learn the invariant causal features for the world model.
>
> **Q3** The quantitative results on iGibson look promising but it is unclear whether the other contrastive learning techniques (like RAD or CURL) use the same data augmentations.
> > **A3** We used official code for all models and data augmentations are similar except CURL proposed to use random-crop but not the color-jitter. We thought it's not necessary to modify their official code as we are additionally using texture randomization for all models anyway. Further, RAD shows that random-crop is better on separating relevant information (ref. Figure 4 from RAD paper)
>
> **Q4** Relatedly, is texture randomization used here? If so, is it used by all contrastive learning based techniques?
> > **A4** Yes, we have used texture randomization features for all the models.
>
> **Q5** the authors point out (in a couple of places) that their technique learns invariant causal features of the environment. However, this really depends crucially on what interventions are available; if the interventions you have do not cover all of the spurious variables then contrastive learning over data augmentations cannot learn causal features (it will learn a combination of causal and spurious). And in fact there is really nothing special about the technique that allows it to learn these causal features; it is the data augmentations (interventions), which are provided by us (the humans) that allows learning these causal features.
> > **A5** We agree that the quality of the causal features is intervention depended. In fact, we have shown that depth is a better auxiliary task than simple RGB reconstruction. Moreover, in this work we show that even the common data augmentation techniques can be used as interventions and help to learn better features, as shown by significantly better results. In some cases our model achieves 17 or more times better results than the current state-of-the-art model. We hope our work will motivate the community to design even better interventions to learn the true causal features or relevant features only, using our framework.
>
> **Q6** Do the augmentations change all the spurious variables?
> > **A6** No, for example our current intervention can't separate the relevant scene-part from the irrelevant scene-part of the observation. Thank you for pointing out this very important aspect of the causal interventions.
>
> **Q7** The authors texture randomization "action replay". This seemed rather confusing to me. How is action replay and texture randomization the same thing?
> > **A7** In our paper, action replay means robot chooses same action sequences after the texture randomization.
>
> **Q8** Do other techniques use the same augmentations? Especially texture randomization?
> > **A8** Yes, all the techniques uses same texture randomization. More on the data augmentations is answered on #A3.
>
> **Q9** For sim-to-real results, do all techniques use texture randomization? Otherwise the comparison is not fair.
> > **A9** Yes, they all used texture randomization features.
>
> **Q10** It'd be nice to have std devs with quantitative results.
> > **A10** We didn't include for two reasons, i) space limit of the paper, ii) for iGibson and Gibson experiments we run for 3 seeds. However, we promise to include all the details in the appendix.

---

> ### Author Response · Authors · 2022-11-15
> **Addressing your concerns and summary of changes (2 of 2)**
>
> **Q11** Please improve table captions. For example, table 4 caption should mention what abbreviations mean?
> > **A11** Improved in the new version, thank you.
>
> **Q12** There are other self-supervised approaches for RL slike SpR (Data-Efficient Reinforcement Learning with Self-Predictive Representations). It'd be nice to compare to them.
> > **A12** Given the current time limit we couldn't finish this but will include this as a future work.
>
> **Q13** Typos
> > **A13** Corrected in the new version and thank you so much for pointing those out.
>
> **Q14** The choice of depth prediction instead of image is not well motivated. And using contrastive learning in an RL setup is not novel.
> > **A14** Motivation behind intervention invariant auxiliary task i.e. here depth prediction is addressed in #A1. Further, unlike other RL models our proposed method here is different: We learn invariant causal features for the world model. Further, as we mentioned earlier designing an intervention invariant auxiliary task is unique to our paper. Moreover, our strong results further enforce the idea.

---

### Official Review · Reviewer_12bZ · 2022-10-25

**Confidence:** 5
**Correctness:** 3
**Technical Novelty And Significance:** 2
**Empirical Novelty And Significance:** 3
**Recommendation:** 3

**Clarity, Quality, Novelty And Reproducibility:**

There are no issues regarding the clarity or quality of this submission.  In terms of novelty, unsupervised and contrastive representation techniques for world models have been explored before.  As this appears to be the main addition to DreamerV2 (as mentioned in the first section of Section 3.3), the novelty of the proposed approach appears rather stale.  Some examples of such approaches include Paster, Keiran, et al. "BLAST: Latent Dynamics Models from Bootstrapping." Deep RL Workshop NeurIPS 2021. 2021 (https://openreview.net/forum?id=VwA_hKnX_kR). and Luo et al. “Visual Control with Variational Contrastive Dynamics.” BeTR-RL Workshop ICLR 2020. 2020 (http://betr-rl.ml/2020/abs/28/) - and more related works can be found readily through a thorough literature search.  I believe there to be no issues regarding reproducibility in this work.

**Strength And Weaknesses:**

A strength of this paper is its sim-to-real perception task performance on iGibson-to-Gibson, which seems to greatly outperform prior model-free results.

Weaknesses of this paper include its novelty (which will be elucidated upon in the following section).  Method-wise, many benefits arise from unsupervised contrastive learning (not requiring labels, focusing on the features that change rather than the ones that remain constant throughout scenes).  However, these benefits seem to disappear when the authors reintroduce reconstruction, whether it be depth reconstruction when such information is available, or regular RGB reconstruction when it is not (e.g. DMControl).  Rather than terming these reconstruction objectives as auxiliary objectives, it seems to me that in fact the contrastive objective is the auxiliary objective added to a default DreamerV2 (at least in the DMControl task).

**Summary Of The Paper:**

In this paper, the authors essentially learn a DreamerV2 model, where the default reconstruction loss has been replaced by a contrastive loss and an additional auxiliary loss (which is depth reconstruction for datasets with such information available, and reconstruction once again for those without).  The contrastive loss is learned in a similar way to the setup in SimCLR, where two stylistic augmentations of the same frame are encoded and treated as positives (and other frames are treated as negatives).  The main claim is that what is learned from the outcome of this contrastive loss are the causal features, where augmentation interventions would not affect the result.

**Summary Of The Review:**

The overall paper was written clearly, and quite easy to follow.  The authors also demonstrate good results on certain tasks, such as sim-to-real.  However, I believe that method-wise, the benefits are limited with the continued use of some form of reconstruction.  Furthermore the utilization of unsupervised and contrastive losses for world models, which appears to be the main contribution in this work, has been previously explored.  Overall, I recommend that this work revisit its novelty and strengthen its contributions before being considered for acceptance.

---

> ### Author Response · Authors · 2022-11-15
> **Addressing your concerns and summary of changes**
>
> We would like to thank you for highlighting the value of the sim-to-real results. Below, we answered all of your concerns.
>
> **Q1** Benefits (of the proposed model/WMC) seem to disappear when the authors reintroduce reconstruction, whether it be depth reconstruction when such information is available, or regular RGB reconstruction when it is not (e.g. DMControl).
> > **A1** One of the key element of our proposed world model is to `design intervention invariant auxiliary tasks`, which allow to learn the invariant causal features. Specifically, we learn features that are more robust against changes in nuisance variables. This is our contribution in the theory, which is unique to our paper.
> >
> > In our work, we design 2 intervention-invariant auxiliary tasks to verify our proposal: Reconstruction of depth or denoising RGB input from the various data augmentations (so it's not exactly the input reconstruction). Other examples of intervention-invariant auxiliary tasks may include dense semantic segmentation, dense scene flow reconstruction and sparse landmarks detection. We hope ultimately this idea can be a catalyst to invent the extraction of the relevant variables only from the whole observation by designing better intervention-invariant auxiliary tasks. We emphasize, rather than simply viewing our method as a depth or RGB reconstructions/decoders, we show theory to support intervention-invariant causal feature extraction.
>
> **Q2** Similar approaches exist: BLAST and VCD (Visual Control with Variational Contrastive Dynamics)
> > **A2** BLAST is based on DreamerV2 as they remove RGB input reconstruction, stop posterior influence to the prior in the KL loss and is inspired by the BYOL paper. It further uses exponential moving average to learn the target/inference-time observation-encoder. Hence, we would like to argue that BLAST doesn't learn the invariant features, and consequently BLAST's performance will be degraded similarly to the RAD in iGibson-to-Gibson sim-to-real test or other downstream tasks.
> >
> > VCD: Our model has two main components, i) invariant feature learning with contrastive loss; and ii) intervention invariant auxiliary task. We agree with the reviewer that (i) is similar to that of VCD. However, as in #A1, (ii) is a key contribution of the proposed model. In another words, using notation of Table 1, **While "WMC - D" is similar to VCD, the proposed WMC is not. Please note that any results without component (ii) would be worse than WMC as shown in the ablation study.** Probably that is why VCD results are only comparable to DreamerV2 (ref. from the VCD paper- "Out of the 20 tasks, InfoNCE ties with it on 8, outperforms it on 4 and underperforms on 8. Triplet ties it on 9, outperforms it on 2 and underperforms on 9.") while WMC is two to three times better than WMC - D + I (the closest model to VCD) (ref. Table 4 in the paper) and 17 or more times better than DreamerV2 (ref. Table 1).
>
> **Q3** Weaknesses of this paper include its novelty.
> > **A3** We hope #A1 and #A2 addressed the novelty issue. Further, since the better performance of the strong data augmentation papers (RAD and others) questioned the benefit of world model and contrastive loss based methods, in this paper we clearly show that the performance of strong data augmentation based methods significantly degrades on out-of-distribution tests, where world model and contrastive learning based methods excel. Hence, we believe that our sim-to-real result is a good contribution on its own.
>
> We hope this answers your concerns and suggestions. If you still have more questions before you feel confident to improve your rating of the paper, please let us know. Once again we really appreciate your time and thank you once again.

---

### Official Review · Reviewer_tiQV · 2022-10-25

**Confidence:** 4
**Correctness:** 4
**Technical Novelty And Significance:** 3
**Empirical Novelty And Significance:** 3
**Recommendation:** 6

**Clarity, Quality, Novelty And Reproducibility:**

The paper has a good structure and quality except for describing some notations/naming conventions in the experiments section. Regarding novelty, it proposes a data augmentation approach to representation learning in world models, which is a very promising direction. The authors of the paper clearly describe the implementation details to be straightforward to reproduce the results.

**Strength And Weaknesses:**

Strengths
- Well-written and organized
- Related work section gives a good comparison between the proposed model and previous work
- Performed experiments for benchmarking are extensive and suitable
- Ablation study is well-designed and confirms the proposed methods

Weaknesses
- Small typos in the text (see questions and suggestions)
- Content of the tables is not always easy to follow (see questions and suggestions)
- Hyperparameters are the same in all experiments

Questions and suggestions for the authors (section by section):
Related work

Q1: In paragraph Sample efficiency: Not the correct paper of the Dreamer method is referenced by the authors when they write about the Dreamer method by Hafner et al. that was beaten by the CURL technique. The referenced paper didn’t even exist that time.

Their reference:
Danijar Hafner, Timothy P Lillicrap, Mohammad Norouzi, and Jimmy Ba. Mastering atari with
discrete world models. In International Conference on Learning Representations, 2021.

Correct reference:
Danijar Hafner, Timothy Lillicrap, Jimmy Ba, and Mohammad Norouzi. Dream to control: Learning
behaviors by latent imagination. In International Conference on Learning Representations, 2020.

+ the style of the title “Sample efficiency” could be bold and not italic to match the other paragraph titles

Proposed model
Q2: In Section 3.2 World Model you describe briefly the controller in the first paragraph: “The controller maximizes the action probability using an actor critic approach”. I found this sentence confusing because these methods aim to maximize the expected reward.

Experiments section
Q3: The meaning of the letters (I, D, AR) should be clearly presented in the first experiments to be easier to follow, they can be only known from the Ablation study, which is at the very end of the section.

Q4: Table 1 and Table 3 (probably typos):  I believe it is a typo in Table 1.: the last but one method in the 500k steps is “Dreamer - I + D + DA”, all the others are DreamerV2. Is it DreamerV2 too? In Table 3. the method called “Dreamer” is the same DreamerV2 as in the other experiments or the earlier Dreamer version of Hafner et al.? + The headers could be more consistent in Table 3.: “100k Steps” and “500K step scores”.
Q5: Table 3: Could you describe what are exactly the presented results in these environments? Are they the obtained rewards?
Q6: Table 1, 2 and 4: It would be good to present the confidence of these models by reporting the standard deviations of SR and SPL.
Q7: Table 1 and Table 4: What is the meaning of the column “Total”?

Q8: The exact hyperparameters are usually very task specific. What is the reason for using the same hyperparameters in these diverse environments (iGibson, DMControl)? Have you done any hyperparameter tuning?

Q9: Ablation study: The authors state that their WMC model performs better in 5 out of 6 environments. However, it is 3/6 in the case of 100k steps and 2/6 in 500k steps.


**Summary Of The Paper:**

This paper provides a novel way towards achieving better representation learning in world models, which is essential to robust policy learning, by unsupervised causal representation learning. The authors of the paper propose depth prediction and data augmentation techniques to reach this goal.  Their model is called World Model with invariant Causal features (WMC). Some of the main contributions and findings of the paper include: 1) Depth reconstruction to achieve better representations. 2) Data augmentation on RGB image input as intervention for contrastive learning. 3) They allow training in a single environment leading to sample efficiency, 4) They also allow tackling training bias in the model. 5) Good results in out-of-distribution generalization in iGibson 1.0 (navigation task) and sim-to-real transfer in iGibson-to-Gibson. 6) Wider applicability of proposed model without depth reconstruction is showed by the DMControl suite experiments


**Summary Of The Review:**

The paper is well-written and presents a novel approach to representation learning problems in world models with extensive experimentation results. Addressing the highlighted minor weaknesses would make the paper stronger. I recommend acceptance.

---

> ### Author Response · Authors · 2022-11-15
> **Addressing your concerns and summary of changes**
>
> We would like to thank you for appreciating our approach. We address your comments below and in our revised submission of the paper, and we hope this gives you the confidence to support our paper with a higher score.
>
> **Q1** a) In paragraph Sample efficiency: Not the correct paper of the Dreamer method is referenced by the authors... b) Also, the font style of the heading "Sample Efficiency."
> > **A1** Both corrected, thank you for pointing these out.
>
> **Q2**  In Section 3.2 World Model you describe briefly the controller in the first paragraph: “The controller maximizes the action probability using an actor critic approach”. I found this sentence confusing because these methods aim to maximize the expected reward.
> > **A2** We meant: The controller maximizes the probability of the action which yields the highest total reward. This is now rephrased.
>
> **Q3** The meaning of the letters (I, D, AR) should be clearly presented in the first experiments to be easier to follow, they can be only known from the Ablation study, which is at the very end of the section.
> > **A3** We now include the meaning of I, D, AR in the caption of Table 1 for clarity.
>
> **Q4**  Table 1 and Table 3 (probably typos): I believe it is a typo in Table 1.: the last but one method in the 500k steps is “Dreamer - I + D + DA”, all the others are DreamerV2. Is it DreamerV2 too? In Table 3. the method called “Dreamer” is the same DreamerV2 as in the other experiments or the earlier Dreamer version of Hafner et al.? +
> The headers could be more consistent in Table 3.: “100k Steps” and “500K step scores”.
> > **A4** It is the same DreamerV2 in Table 1, which is corrected now. And, in Table 3, it's the earlier Dreamer version only (Hafner et al., 2020). Further, following the suggestions the headers are made consistent now.
>
> **Q5** Table 3: Could you describe what are exactly the presented results in these environments? Are they the obtained rewards?
> > **A5** Yes, they are obtained rewards. Results are reported as averages across 10 seeds.
>
> **Q6** Table 1, 2 and 4: It would be good to present the confidence of these models by reporting the standard deviations of SR and SPL.
> > **A6** Due to the high demand of computational time, we reported the Gibson and iGibson results as average across 3 seeds. So, we didn't include the standard deviation. However, we are happy to include if you advise so with just 3 runs. Otherwise, we are going to release all the individual values in the appendix.
>
> **Q7** Table 1 and Table 4: What is the meaning of the column “Total”?
> > **A7** Total shows the average results across the environments for the given method (row-wise). We now updated this.
>
> **Q8** The exact hyperparameters are usually very task specific. What is the reason for using the same hyperparameters in these diverse environments (iGibson, DMControl)? Have you done any hyperparameter tuning?
> > **A8** We picked the published hyperparameters from the respective model/paper and haven't done any hyperparameter tuning including WMC for two reasons- i) robustness/wider-applicability of the models, ii) computational cost.
>
> **Q9** Ablation study: The authors state that their WMC model performs better in 5 out of 6 environments. However, it is 3/6 in the case of 100k steps and 2/6 in 500k steps.
> > **A9** Note, the referenced line is about Table 1 and Table 4 (we meant 5 out of 6 'experiments' (not 'environments')). Specifically, we compare the results of "DreamerV2 - I + D + DA" from Table 1 vs "WMC - AR" from Table 4. This comparison highlights the importance of the contrastive loss in MBRL.

---

> > ### Comment · Reviewer_tiQV · 2022-11-21
> > **Reviewer Response**
> >
> > We appreciate the time the authors took to make the necessary changes, especially in the experiments section which make the paper easier to follow.
> >
> > Regarding Table 1, 2 and 4, we believe it is sufficient if you report the standard deviations with 3 runs in the Appendix.
> >
> > We also advise the authors to check again the columns of Env. Avg. in Table 1 and 4, because we noticed that the WMC-AR-D with 100k has clearly an incorrect Success Rate value or the individual elements need to be corrected.

---

> > > ### Author Response · Authors · 2022-11-21
> > > **Clarification on Env Avg values in tables 1,4**
> > >
> > > This is because, in the Env Avg column of Tables 1,4 we have taken the average of the success rates/SPLs of individual environments before rounding them off. Thanks for pointing it out, we will change them to average of the rounded values so that they are consistent with other columns.

---

### Author Response · Authors · 2022-11-18
**Summary of Response**

**Importance of our findings.** Contrary to the recent findings of RAD and DrQ which advocated data augmentation is sufficient for sample efficiency and generalization, we have shown that Model-Based RL with invariant causal features are more suitable and further improve sample efficiency and generalization to downstream tasks such as sim-to-real transfer. Our results on sim-to-real not only provide a significant new state of the art, but our findings also present a new framework for auxilary task driven intervention for improved invariant causal feature learning in Model-based RL. We believe, this contribution is important to be shared with the RL community.

**Novelty.** In this paper we propose to learn a world model with invariant causal features using i) contrastive learning, and ii) intervention invariant auxiliary tasks. A contrastive learning loss implicitly enforces the invariance. Our intervention-invariant auxiliary task explicitly forces the invariance, which is one of the main contribution of our work. Contrary to the peception of some reviews, our focus is not the depth auxilary task itself, but rather its use as *intervention-invariant* auxilary task in Model-Based RL. Other intervention-invariant auxilary tasks may include semantic segmentation or scene flow to name a few. Which auxilary to use is task specific and dependent on available data. Note also, in our experiements with the DeepMind control suite where depth is not available, we use simple image denoising as auxilary task for RGB data augmentation. We agree, neither auxilary task fully seperates causal features, but both provide a better guide to the Model-based RL. In our ablation study we show significant improvements through our intervention-invariant auxilary task framework. Further indication of invariant causal feature learning is much improved generalization in our out-of-distribution and sim-to-real evaluation. Finally we emphasize, our framework is general, and future work may introduce better auxilary tasks for better causal feature discovery.

**Thanks.** We thank the reviewers for their time and helpful comments; we appreciate that all reviewers valued the significantly improved results on the iGibson-to-Gibson sim-to-real transfer task and our thorough ablation study; and we believe the simplicity of our causal feature learning framework will lead to fruitful future research in the community.

---

### Decision · Program_Chairs · 2023-01-20

**Decision:**

Reject

**Justification For Why Not Higher Score:**

As pointed the above, the novelty is too limited to be accepted.

**Justification For Why Not Lower Score:**

N/A

**Metareview: Summary, Strengths And Weaknesses:**

This paper proposes a method for world model learning. It is based on DreamerV2 but proposes to remove image reconstruction, add depth prediction and contrastive auxiliary loss. The proposed model is tested in iGibson dataset and provides a good results.

The main strength of the paper is to demonstrate that the proposed method is working well on a specific benchmark, iGibson.

The main weakness is the novelty. Introducing contrastive loss to learn world model is not new, e.g., see, BLAST (Paster et. al 2021) or Temporal Predictive Coding (TPC) (Nguyen et. al. 2021). It is not interesting to remove the reconstruction loss but to reconstruct the depth image. The depth image is not general modality, and also can be seen as a type of image. The good performance is also limited only to the iGibson dataset. Also, some reviewers found that the writing is not clear or matured, and I also agree.